# Modbus Access Control System Based on SSI over Hyperledger Fabric Blockchain

**DOI:** 10.3390/s21165438

**Published:** 2021-08-12

**Authors:** Santiago Figueroa-Lorenzo, Javier Añorga Benito, Saioa Arrizabalaga

**Affiliations:** 1CEIT-Basque Research and Technology Alliance (BRTA), Manuel Lardizabal 15, 20018 Donostia/San Sebastián, Spain; jabenito@ceit.es (J.A.B.); sarrizabalaga@ceit.es (S.A.); 2School of Engineering, University of Navarra, Tecnun, Manuel Lardizabal 13, 20018 Donostia/San Sebastián, Spain

**Keywords:** Modbus, IIoT, blockchain, hyperledger fabric blockchain, self sovereign identity, access control system, authorization, authentication

## Abstract

Security is the main challenge of the Modbus IIoT protocol. The systems designed to provide security involve solutions that manage identity based on a centralized approach by introducing a single point of failure and with an ad hoc model for an organization, which handicaps the solution scalability. Our manuscript proposes a solution based on self-sovereign identity over hyperledger fabric blockchain, promoting a decentralized identity from which both authentication and authorization are performed on-chain. The implementation of the system promotes not only Modbus security, but also aims to ensure the simplicity, compatibility and interoperability claimed by Modbus.

## 1. Introduction

Modbus became one of the most widely used protocols in Industrial Internet of Things (IIoT) environments and nowadays it is implemented by hundreds of different vendors and thousands of devices [1]. Modbus enables the data exchange between different parts of the industrial process, not only Programable Logic Controller (PLCs) and field devices but also PLCs and Supervisory Control and Data Acquisition (SCADAs). The three main features that enable Modbus success are simplicity, compatibility and interoperability. However, security is its main deficiency. In this regard, vulnerabilities have been detected both in design and configuration as well as in implementation, which are exploited using different attack patterns [1]. Hence, several solutions add security layers while trying to keep it simple, compatible, and interoperable. Some of these solutions belong to identity and access management domains. For instance, this approach provides Modbus with trusted identity and mutual authentication based on X.509 certificates, as well as secure authorization based on a Role-based Access Control (RBAC) [2]. These types of common solutions, based on centralized systems, are optimal in single-organization environments, being the scalability their main drawback (i.e., solution feasibility is affected when multiple organizations are involved). In this sense, the authors of [3] justify scalability problems in terms of PKI infrastructure costs in Smart Grid environments, while the authors of [4] justify it based on scenarios such as certificate revocation. However, a concern with central authorities controlling identities is that if they are compromised in some way, those identities can be used in malicious ways: e.g., a hack of Dutch Certificate Authority allowed supposedly secure encrypted data going across the internet to be intercepted and accessed by hackers [5]. The decentralization promoted by blockchain emerges as a solution to both scalability and centralized identity challenges, since it introduces an environment with multiple organizations, while enabling the devices’ self-custody of identifiers and credentials. In that sense, different works introduce innovative solutions for identity and access control management based on blockchain for IoT environments [6,7]. However, none of them natively support the use of Self-Sovereign Identity (SSI) for decentralized authentication and authorization and less for an IIoT environment represented by Modbus. The manuscript’s contribution is the design and implementation of a Modbus Application Protocol Secured based on Self-Sovereign Identity (mbapSSI) over Hyperledger Fabric Blockchain (HFB) that ensures not only on-chain authentication and authorization but also the other phases of an access control system: identification, auditing, and accountability, constituting an identity and access management system. To verify the feasibility of mbapSSI, performance and scalability analyses are carried out in constrained Modbus-IIoT environments. The rest of this manuscript is structured as follows: Section 2 introduces the context related to blockchain-SSI as an identity and access management system, examining then, the application of SSI in the context IoT and IIoT as part of the related work. Section 3 provides a background including Modbus protocol, SSI, and HFB, i.e., mbapSSI enabling technologies. The design and implementation of our system are described in Section 4. Section 5 describes the testbed and conducted experiments to determine the feasibility of the proposal. Section 6 discusses the results of conducted experiments. Finally, the conclusions of the manuscript are included in Section 7.

## 2. Related Work

Enabling the aforementioned features for mbapSSI implies going beyond an access control system or an identity management system. For this reason, this section discusses the integration of both concepts. Based on the Identity and Access Management (IAM) Framework [8], the scientific community, governments and enterprises establish blockchain as an enabling technology for both Identity Management (IM) and Access Management (AM). In the identity management domain, Blockchain Identity Management Systems (BIMS) are considered as an emerging technology, which differs of Traditional Identity Management System (TIMS) since blockchain enables the custody of identity information, while TIMS stores credentials (e.g., password) about users and devices which they interact [9]. Currently, governments and organizations are working hard to create guidance on BIMS. Thus, some countries, such as Estonia, are experimenting with BIMS for electronic medical records [10]. In the access management domain, blockchain technology has been included as part of several scenarios that involve all phases of access control systems: identification, authentication, authorization, auditing and accountability [11]. At this point, the SSI paradigm, does not need blockchain to be implemented. Nevertheless, blockchain offers benefits that can be exploited by an integrated SSI-blockchain solution: blockchain can be used as a distributed ledger to establish immutable records of lifecycle events for globally unique decentralized identifiers (DIDs). In addition, SSI-blockchain integration can be the enabler for identity and access management. Although the authors of [12] classify SSI only as an emerging identity management system, the fact that SSI ensures not only authentication, but also authorization [13], implies that it is also involved in the context of access management. Once having discussed that SSI is suitable for IAM, we will focus on the related work in the application of SSI in the context of IoT and IIoT due to the recent integration of both concepts.

Thus, the following manuscript [14] contrasts existing identity approaches, such as digital certificates, with open standards for SSI: decentralized identifiers (DIDs) and verifiable credentials. The same work also analyzes the advantages and challenges of both standards for ensuring authentication in IoT environments, although these are only proposals that do not result in the design or implementation of a specific system but are part of future research lines. Additionally, the work in [15] compares SSI-blockchain solutions such as Sovrin, UPort and Jolocom, while promoting the integration of SSI in IoT architectures, enabling comparisons with OpenID Connect. However, again, this work is only a very useful proposal (not implementation) in terms of background and analysis of possible use cases. Furthermore, the Sovrin approach indicates, from a non-technical perspective, how SSI addresses main IoT challenges such as identification, authentication, authorization and auditing, while ensuring data privacy and data integrity over a secure channel [16]. However, this manuscript also focuses on proposals rather than specific designs or implementations of SSI. Finally, the authors of [17] present a novel solution for IoT device IM based on SSI and backed by the security offered by the IoTA Tangle DLT. Although the authors even present a practical use case focused on car rental industry, there are no technical details neither performance analysis of the implementation of the solution. Despite the progress described above, numerous challenges face SSI in both the IoT and IIoT contexts, including authentication and authorization in machine-to-machine (M2M) environments [13]. Therefore, mbapSSI aims to address these challenges, not only from a theoretical design in a representative IIoT use case, but also from the implementation and feasibility analysis point of view, trying to solve the mentioned gaps. Thus, mbapSSI does not only guarantee the decentralized identity management that blockchain promotes, but also includes the design and implementation of authentication and authorization on-chain. The mbapSSI approach consists of authorizing the use of a resource and giving access to a Modbus service, putting in value the role of blockchain technology for SSI, as a trust mechanism that allows controlling access to data, allowing the Modbus device to decide with whom, when and how it shares information.

## 3. Background

The three main concepts of mbapSSI are discussed in this section: Modbus protocol, SSI and Hyperledger Fabric Blockchain.

### 3.1. Modbus Protocol

Modbus is an Industrial Internet of Things (IIoT) protocol with applications in scenarios such as building automation or energy management systems [1]. Modbus is useful for the management and control of industrial devices such as Programable Logic Controller (PLCs), Supervisory Control and Data Acquisition (SCADAs), sensors and actuators. The three most common versions of the protocol are Modbus ASCII, Modbus RTU and Modbus TCP/IP, where the first two use serial line interfaces and are associated with deterministic transmissions. Modbus TCP/IP arises with the objective of solving requirements such as the limit to 240 devices per network, however, the non-deterministic nature of TCP/IP means that it is common to see this version of Modbus in interactions between supervisory level and field level devices (e.g., SCADAs and PLCs) rather than interactions between field devices with each other such as a PLC and a sensor. From performance perspective, Modbus-IIoT environments can be considered as constrained scenarios, which can result in latencies of up to 100 ms according to [18]. It is well known that the major shortcomings of Modbus are in the area of security, for this reason references such as [19] present an extensive taxonomy of attacks for both Modbus serial and Modbus TCP. Considering that standard Modbus protocol cannot be secured, e.g., providing authentication to the Modbus frame, whatever security layer provided, it must not limit the simplicity, compatibility and interoperability of the Modbus protocol.

### 3.2. Self-Sovereign Identity (SSI)

SSI follows the basic premise that people should control their own identity with regard to relationships and interactions with other people, organizations, and things [20]. However, the evolution of the concept of SSI extends the control of their identity to the context of things and machines, as mentioned in Section 2. The principles guiding SSI have been presented in the past by Christopher Allen [21], however, approaches that extend these principles have been analyzed by the authors of [12], including concepts such as (1) sovereignty, (2) data access control, (3) data storage control, (4) security, (5) privacy, (6) flexibility, (7) accessibility and (8) availability.

The SSI ecosystem assumes three key roles (issuer, holder, and verifier) in addition to a verifiable data and status registry (VDSR). An issuer creates and issues credentials to a holder. A holder receives credentials from an issuer, holds them and when required, it shares these credentials with a verifier. A verifier receives and verifies credentials presented by a holder. A VDSR is a trusted mediator to manage and verify relevant information, e.g., identifiers. The verifiable credentials’ specification defines examples of VDSR such as a trusted database, decentralized databases, and distributed ledgers [22].

SSI integrates some standards such as Verifiable Credential (VC) [22] and Decentralized IDentifier (DID) [23], which are proposed to create a cryptographically verifiable digital identity that is fully controlled by its owner [24]. A VC is an attestation of qualification issued by a third party (e.g., issuer) with de facto authority to an entity (e.g., holder) [24]. In this regard, VCs are JSON documents constructed and digitally signed by an authority (e.g., issuer) which includes possible correlating values such as a holder identifier, the signature value and the claim value. The DID is a permanent, universally unique identifier and cannot be taken away from its owner who owns the associated private key, which is completely different from other identifiers such as an IP address and domain name [24]. Two other important concepts associated with DID should be analyzed: DID Method and DID Document. DID Methods define the types or classes of DIDs and represents the second part of the DID format. Several types of DID Methods are set out in [25], including Ledger-based DIDs, Ledger Middleware, Peer DIDs, Static DIDs and Alternative DIDs. Since Ledger-based DIDs are the most commonly used, some examples are shown next: btcr (Bitcoin), ethr (Ethereum), and sov (Sovrin). DID Document is a data structure that contains basic information that is needed to connect with a subject (e.g., a verifier). The DID Document is resolved through a DID and consumed by digital identity applications (e.g., wallet), so that each DID has exactly one DID Document associated with it [25].

### 3.3. Hyperledger Fabric Blockchain

HFB is a private and permissioned distributed ledger and smart contract framework maintained by Linux Foundation [26]. Considering that HFB is pivotal to our proposal, we will justify its selection as a VDSR of our SSI proposal from three perspectives. First, the use of HFB as part of the IAM context, highlighting its suitability in SSI scenarios. Second, the justification of using HFB over other SSI implementations. Thirdly, from a performance perspective given our focus related to IIoT environments.

In the identity management domain, HFB is involved in several use cases. For instance, the work in [27] uses HFB as a Certification Control System to enforce Certificate Signing Request (CSR) validation. In addition, HFB is also used to provide transparency for better Certificate Authority (CA) accountability [28]. In the access management domain, there are also several use cases, including HFB as part of SSI systems. Thus, several approaches use Access Control Systems based on HFB to ensure phases such as identification, authentication, authorization and accountability in IIoT environments [15,29,30]. Moreover, in the context of SSI, although the scientific literature offers few examples involving HFB in SSI systems, some steps have been taken towards this direction. Thus, the contribution in [31] designs and implements both an SSI and an access management system for smart vehicles, using HFB as a key piece.

References such as [15] analyze SSI solutions including Sovrin, Civic, UPort, Jolocom, Veres One, etc. These solutions mostly implement two types of blockchain technologies: Ethereum and Hyperledger Indy, which will be the objects of our comparison with HFB. Although solutions such as uPort (Ethereum) seem promising [32], on-chain environments are limited from both solidity capabilities and order–execute architecture. In contrast, HFB through the execute–order–validate model eliminates non-determinism and enables standard programming languages such as Golang, which amplifies the on-chain capability of the solution. Additionally, Hyperledger Indy is a purpose-built ledger for Identity, which can be considered a public-permissioned blockchain [33]. However, since Indy does not have the ability to run smart contracts, authentication and authorization are based on information stored in the ledger using static and predefined rules. From the perspective of mbapSSI, the on-chain dynamism provided by a chaincode, allowing complex computational operations such as on-chain signature verification, exceeds the performance of Indy, a scenario that implies the need to strengthen the off-chain implementation.

From a performance perspective, according to [34], blockchain addresses challenges defined as key for IIoT environments such as system scalability and interoperability, security and data quality. Particularly, it has been demonstrated the feasibility of an HFB-based access control system in a performance-constrained IIoT environment such as an engine assembly line [35].

## 4. Design and Implementation

This section establishes the design and implementation criteria defined for mbapSSI. For a better understanding of these criteria, we contextualize the background provided for SSI in terms of key concepts for mbapSSI.

### 4.1. Key Concepts for mbapSSI

#### 4.1.1. Modbus End-Devices Analysis for mbapSSI

Section 3.1 mentioned the low latency requirements for a Modbus-IIoT environment. Considering the network architecture recommended by NIST [36], mbapSSI puts the focus on the interaction between field control devices with local human–machine interface (HMI), i.e., TCP/IP connections. In this way, a Modbus client could be visualized as a Scada, while the Modbus server as a PLC. In addition to network communication, Modbus devices store a key pair associated with the DID as part of a wallet.

#### 4.1.2. DID Approach for mbapSSI

Expression (1) shows the Ledger-based DID contextualization for mbapSSI on a three-element identifier separated by a colon: scheme identifier “did”, DID Method identifier and DID method-specific identifier. The approach adopted in mbapSSI for three-parameter definition is based on [25], following the DID syntax based on Augmented Backus-Naur Form (ABNF) Syntax Specification [37] as DID Standard defines [23]. In this regard, the DID Method indicates the ledger used as VDSR, defining as DID work with a specific blockchain. The specific identifier of the DID Method satisfies the property of being unique in the namespace of the DID Method, since it is the SHA-256 hash function of the public key registered in the VDSR, being the entity identified by the DID, the holder of the private key. Additionally, with regard to the DID infrastructure, mbapSSI promotes a key-value database, where key is the DID and value is the DID Document. Thus, we can confirm that the DID adopted by mbapSSI complies with the properties of being permanent, resolvable, cryptographically verifiable and decentralized [38], being therefore a very close approximation to the DID standard by satisfying some of the core properties of the standard.
did:hfb:1fb352353ff51248c5104b407f9c04c3666627cf5a167d693c9fc84b75964e2(1)

Considering that mbapSSI aims to improve the shortcomings of the PKI infrastructure presented for Modbus in terms of scalability, the first objective of the DID will be to integrate it into X509 certificates, to replace the functionality of the Certification Authority (CA) associated with a PKI infrastructure, as a base for the security of Modbus devices in centralized environments according to the Modbus security specifications [39]. The purpose behind not discarding the use of X509 certificates is that it is a mandatory requirement for securing the channel using TLS. In this way, the DID of the Modbus device is recorded as the Subject Alternative Name (SAN) extension within the X.509 certificate [40], allowing the VDSR to become the authority for verifying identity rather than a CA when establishing a secure channel. In this way, when Modbus devices exchange certificates as part of TLS, they will be able to verify the DID through the VDSR. The second objective of DID is to return the associated DID Document when the VDSR is properly queried, thus fulfilling the resolvable property.

#### 4.1.3. DID Document Approach for mbapSSI

Our mbapSSI uses the “resolvable” property of DID to record, in the VDSR, the key-value type for the DID-DID Document relation. Thus, our DID Document contains a context, the authentication mechanism supported, the Modbus service definition, and the digital signature of the entity that creates it. The Modbus service definition is fully compliant with Modbus specifications [41], so it includes the service endpoint, the function code, the starting address and the offset. As expression (2) indicates, service endpoint is constituted by the Modbus application protocol secured (mbaps) frame type, as well as IP address and access port. Each of the Modbus services to be exposed must be defined in the DID Document. The DID Document can be constantly updated by adding or removing Modbus services.
“serviceEndpoint”: “mbaps://{}:{}”.format(ipaddress, port)(2)

#### 4.1.4. SSI without VC Approach for mbapSSI

Although in Section 3.2, VCs were defined, their standard indicates that their use is not mandatory [22]. Thus, mbapSSI does not follow the VC model but focuses on complying with the SSI properties established by [12], particularly in data access control, data storage control, security and privacy. In that sense, mbapSSI promotes a decentralized identity that highlights the property of sovereignty in the fact that the Modbus server has the ability to share its data (through a Modbus service) with the authorized Modbus client. In mbapSSI, the Modbus client provisioning on the authorization whitelist is performed through the intervention of a user. It is identified as next objective the design and implementation of an SSI model based on M2M authentication proposed by OAuth in [42], but it is out of the scope of this work.

The authorization whitelist includes as input an out-of-band (OOB) mechanism provided by mbapSSI to associate the client’s public key to a Modbus service included in the DID Document. This authorization whitelist will be updated in the VDSR and constitutes the single point of user intervention.

### 4.2. Overview of mbapSSI

In order to integrate the concepts defined in the Section 4.1, as well as to provide an overview of mbapSSI, Table 1 relates the parts that compose an SSI system with the mbapSSI system phases. In this regard, the registration phase provides the Modbus device (holder) with a decentralized identity, ensuring that this identity is registered in the VDSR. In this phase, the DID has been integrated into the X509 certificate. In the provisioning phase, the Modbus server’s owner (user) matches the Modbus clients with the Modbus services to be used, so that the authorization whitelist is updated. It should be noted that the provisioning phase is unrelated to the others, so it can be repeated at any time. The channel securing phase provides the Modbus devices (both being holders and verifiers as it will be clarified later on), with on-chain authentication, as well as the establishment of a secure TLS channel [43]. The verification phase authorizes the Modbus client (holder) to use a resource, i.e., a Modbus service, through a proof of identity provided by the on-chain query of the authorization whitelist, where the Modbus server (verifier) is involved. Finally, Modbus transactions phase enables the exchange of Modbus frames through a secure channel between Modbus client and server (both holders). The VDSR is involved in all phases of mbapSSI except when Modbus devices transact with each other to ensure both low levels of latency and high throughput.

According to [44], in an SSI context each entity can have multiple roles, e.g., holders can also be verifiers. Extrapolating this concept to a Modbus context throughout the mbapSSI phases, the devices supporting the Modbus protocol, i.e., the Modbus client and the Modbus server, may behave as holders or verifiers depending on the phase requirement. In this regard, we define a Modbus client acting as a holder as C_H_, Modbus client acting as a verifier as C_V_, Modbus server acting as a holder as S_H_ and Modbus server acting as a verifier as S_V_. Table 2 summarizes the behaviors per phase. 

Figure 1 shows an overview of the interaction between parties of the SSI context and Modbus devices of the Modbus context. As Table 2 indicates, the parties involved in the registration phase (C_H_ and S_H_) register their identities in the VDSR (1). Next, Modbus service(s) to be exposed are registered in the VDSR by S_H_ (2). At this point and as part of the provisioning phase, in the step (3), the server’s owner relates Modbus client information to the Modbus service(s) exposed in step (2), and then S_H_ updates the whitelist (4).

The channel securing phase is composed of two steps. On the one hand, C_H_ and S_H_ start to ensure the communication channel via TLS (5). On the other hand, Cv and Sv must verify the identities through the VDSR (6). At this point, C_H_ must provide access proof for services through the VDSR (7), based on the information defined in steps (3–4). Finally, Modbus devices (C_H_ and S_H_) exchange Modbus transactions for authorized services (8).

The correlation between Modbus and SSI converges in mbapSSI, where the three initial phases (Registration, Channel Securing and Verification), excluding the Provisioning phase, constitute the necessary mechanism, based on SSI, to provide a secure communication channel, based on TLS, between a client and a Modbus server. The fourth phase is thus isolated to ensure the simplicity, compatibility, and interoperability of Modbus transactions.

### 4.3. Chaincode Design

The chaincode is the most important piece of mbapSSI. Its strength lies in the fact that complex operations such as signature verification can be performed on-chain, something practically impossible in public-permissionless blockchains such as Ethereum, due to the high transactions cost. Therefore, it is possible to assert that the role of HFB for mbapSSI goes beyond a simple VDSR. From a design perspective, the main feature of the chaincode is the presence of a proxy, which is an approach pioneered by UPort [45], although UPort deploys a new smart contract proxy for each identity. In that sense, the design of the proxy as part of our chaincode adopts the identity verification functionality as well as the routing capability to other functionalities. However, the mbapSSI chaincode does not need to generate a new Smart Contract for each identity but relies on on-chain mechanisms such as signature verification to perform identity verification. Therefore, in our case the identity is stored in the ledger and can be queried from the proxy itself. Figure 2 describes the functionalities of mbapSSI’s chaincode: *Get Identity*, *Set Identity*, *Update Whitelist*, *Get Entity’s Identity*, *Set DID Document* and *Get DID Document*. The default design criterium is that any input to the chaincode from the different entities (C_H_, C_V_, S_H_, and S_V_) must have a DID and payload structure as shown in expression (3). For payload processing by the chaincode, it must meet two conditions: (1) the sender identity must be registered and (2) the payload must be signed by the sender.
(3)
{did: “did:hfb:1fb352353ff51248c5104b407f9c04c3666627cf5a167d693c9fc84b75964e2”,payload: “eyJhbGdvcml0aG0iOiJQUzI1NiIsImFsZyI6IlBTMjU2In0.eyJmdW5jdGlvbiI6ImNyZWF0ZVNlbGZJZGVudG”}

To enable these conditions, a proxy structure has been designed that basically verifies both identity and signature, and then deserializes the payload in order to obtain the service to be executed. For this purpose, the proxy relies on other structures, such as the signature verifier and the identity registry, which are involved in the other functionalities except for the *Set Identity* where the payload of expression (3) only contains the public key of the entity. In this regard, the *Get Identity* functionality is only executed within the chaincode and is called several times in all functionalities, in order to retrieve the public key for an entity with a registered identity. In addition, the *Update Whitelist* functionality stores the set of relations of the public key with the Modbus service index in the DID Document. On the other hand, the *Get Entity’s Identity* functionality returns to off-chain side the public key associated to an identity. Additionally, the *Set DID Document* functionality allows to create and later update the DID Document. Finally, the *Get DID Document* functionality allows to retrieve the DID document, for which, a JSON Web Token (JWT) must be included to prove not only the identity of the client, but also the validity of the authorization given to use the service defined in the whitelist. Thus, the public key associated to the C_H_ identity is used to retrieve the Modbus service(s) index(es) from the whitelist, returning to off-chain side, the DID Document associated to the C_H_-related services in provisioning phase.

### 4.4. Implementation

The mbapSSI system is composed of the aforementioned chaincode and a Software Development Kit (SDK) to enable the integration and interaction with HFB network. Thus, while the chaincode manages the interactions in the on-chain side, the mbapSSI-SDK manages the interactions in the off-chain side. This SDK, written in python is imported into Modbus devices, i.e., client and server, to expose all functionalities required to interact with the HFB network, enabling these devices to become C_H_, C_V_, S_H_, and S_V_. However, mbapSSI-SDK imports three key pieces of code for the different interactions throughout the mbapSSI lifecycle. Firstly, mbapSSI-SDK makes use of the HFB SDK for python to invoke (write operations) and query (read operations) transactions on HFB [46]. Secondly, mbapSSI-SDK requires the use of the python SSL library, to build TLS/SSL wrapper for socket objects [47]. Finally, to transact Modbus frames from clients and servers, mbapSSI-SDK imports the pymodbus library [48], a full Modbus protocol coded in python. Since python is not one of the official HFB languages, it makes it necessary to use the HFB version to 1.4.x since the python HFB-SDK updates are delayed for languages such as JavaScript. Thus, to run mbapSSI, both the client and server must include python 3.8 and mbapSSI-SDK into docker images.

The Figure 3 shows a flowchart, which details interactions between entities (C_H_, C_V_, S_H,_ and S_V_), including the information exchanged with the VDSR, corresponding to the phases of mbapSSI. Although the figure uses generic names for easy understanding of the functionalities (e.g., *Request Did Document*), each interaction with the blockchain follows the format defined by expression (3).

Now, as part of the registration phase both S_H_ and C_H_ create and register their DID in the VDSR via the *setIdentity* method. Next, S_H_ is able to create the DID Document via the *setDidDocument* method, although it may be updated each time the server has to expose a new service using the same method. The only requirement for the provisioning phase to take place is that S_H_ has registered the DID Document, so that the set of services to be exposed are selectable by the server’s owner. At this point, the user relates the public key of the Modbus client (C_H_) and the service(s) he wants to access, using an OOB mechanism such as a graphical interface (GUI). This relationship is updated in the blockchain via the *updWhiteList* method. As part of channel securing phase, the C_H_ request TLS connection and after some steps of TLS specification [43] S_H_ sends its certificate (X.509), where, according to Section 4.1.2, both C_H_ and S_H_ have included DID as part of the certificate’s SAN extension. In this way, both client and server, with verifier role, i.e., C_V_ and S_V_ will be able to verify each other’s identity (via *getIdentity* method), since blockchain (VDSR) plays the role of CA. Once the mutual TLS is completed, the channel is secured. As part of the verification phase, C_H_ computes off-chain JWT, which is included as argument of *getDidDoc* method. Thus, the VDSR requests on-chain, the authorization whitelist returning the service(s) enabled to C_H_. In this way, S_H_ authorizes the use of a resource, granting access to a Modbus service, without the need for interaction with C_H_, putting in value the role of blockchain technology for SSI, as a trusted mechanism that allows regulating the access to the S_H_ data, enabling it to decide with whom and when it shares its information. C_H_ and S_H_ are ready to transact Modbus frames.

In order to clarify the off-chain process, when the client requests the DID Document within the verification phase, the mbapSSI-SDK will be used in the off-chain side, in or der to create an object containing the *getDidDoc* method in JSON format. This method uses as arguments a JWT and the server’s DID. This plain text object is firstly encoded in base 64, constituting the first part of the payload of expression (3) which uses the dot as a delimiter. Secondly, the same object is signed and then encoded in base 64, constituting the second part after the delimiter. At this point, the client DID and payload are sent as transaction arguments.

## 5. Performance Evaluation

This section focuses on the evaluation of mbapSSI in terms of performance to determine its feasibility. The performance tests aim at isolating the first three phases from the fourth phase. This is because the value of mbapSSI lies in the fact that the three initial phases, excluding the provisioning phase, constitute the necessary mechanism (based on SSI), to provide a secure communication channel (based on TLS) between a client and a Modbus server. Therefore, the fourth phase is abstracted from the others to ensure the simplicity, compatibility and interoperability of Modbus transactions.

Three performance metrics (processing time, latency, and throughput) are used for the performance evaluation. Processing time represents the time required for a process (e.g., a piece of code) to handle a given request. Latency is a measure of round-trip delay. Throughput constitutes the ratio of the total transactions committed into the ledger in 1 s (not all the sent transactions are always committed in the same second and are pooled in the ordering node). The three metrics are evaluated in different ways, so processing time is used to measure the performance of the mbapSSI phases: registration, channel securing and verification (hereinafter referred to as “three-phases”); latency is used to measure the performance of Modbus transactions; throughput is used to measure the performance of mbapSSI interactions with HFB. Similarly, scalability is another important point in the evaluation of mbapSSI performance, to determine the behavior of mbapSSI due to the increase of organizations and therefore of requests, an n:1 connection pattern is followed, i.e., “n” clients to “1” server. The provisioning phase is not measured since it constitutes an OOB mechanism isolated from the other phases. The remainder of this section examines the testbed used and the experiments conducted to demonstrate the feasibility of mbapSSI.

### 5.1. Testbed Description

Figure 4a shows the test network to be deployed. It is composed of organizations, each of which contains a set of entities such as Modbus entity, orderer, peer_0_.org and CA. Each organization constitutes a module and as needed new modules are added to the testbed, which provides scalability. To achieve integration within a module the deployment flexibility offered by the Docker infrastructure is used so that all elements represented for an organization (Modbus entity, orderer, peer_0_.org and CA) are deployed from Docker images. These containers are integrated into Docker Swarm worker nodes. In this way each organization constitutes a worker node which will be orchestrated using Docker swarm.

The default testbed would be two modules (organizations) including 1 Modbus-client (e.g., Modbus-client_1_) and 1 Modbus-server. To obtain accurate results, an isolation environment is needed, which is provided by an Amazon Elastic Compute Cloud (EC2) instance, whose main feature is scalable performance from a basic level of CPU and memory. The mbapSSI testbed included the deployment of a t2.2xlarge instance, which contains 8 intel AVXT CPUs (3.0 GHz) and 32 GB of memory [49]. It should be noted that the main decision criteria for the deployment of a testbed with these characteristics has been that the HFB-network is deployed using Raft as an ordering service network instead Kafka or Solo since it introduces less centralization [35].

### 5.2. Experiments Conducted

This section describes the experiments to be performed to demonstrate the feasibility of mbapSSI and determine the overhead that mbapSSI introduces to the HFB network. For this purpose, four types of experiments are conducted: Section 5.2.1 describes the experiment that measures the processing time of the three-phases and the latency of the Modbus transaction phase with a client–server 1:1 ratio; Section 5.2.2 describes the experiment that measures the behavior of the processing times of the three-phases of mbapSSI considering the scaling of the client–server ratios as follows: 4:1, 8:1, 16:1 and 32:1; Section 5.2.3 describes the experiment measuring the latency behavior of the Modbus transaction phase of mbapSSI in isolation, considering the scaling of client–server ratios as follows: 4:1, 8:1, 16:1 and 32:1; Section 5.2.4 describes the experiments to determine how many transactions per second are supported by the deployed HFB network.

#### 5.2.1. Performance of mbapSSI Phases at 1:1 Ratio

This experiment focuses on measuring the processing time of the three-phases of mbapSSI that provide a decentralized identity, a secure channel, and the authorization to use the Modbus resource, as well as the latency of the Modbus transaction phase. From the testbed of Figure 4a, this experiment requires the deployment of two organizations, one holding the client and the other holding the server. In order to measure the processing time, since mbapSSI is based on python 3.8, we use the “process_time” tool of the time library. Expressions (4) and (5) are the ways of using the tool, i.e., inserting them in the code at the beginning and at the end of each phase, respectively. To determine the latency of the Modbus transactions, a performance test was designed based on the flowchart shown in Figure 4b, which represents a piece of code written in Python running on each Modbus client, which simultaneously opens 10 connections (threads) to the server, performing 1000 requests for each of the threads, collecting latency introduced by each of these connections, so that the maximum and minimum value of this metric can be evaluated.
start = time.process_time (),(4)
end = time.process_time ()-start,(5)

#### 5.2.2. Performance of the First Three Phases of mbapSSI Based on n:1 Ratio

This experiment aims to measure the behavior of the processing time of the three-phases of mbapSSI under the stimulus of scaling the number of Modbus clients connected simultaneously in ratios of 4:1, 8:1, 16:1 and 32:1, following the architecture of Figure 4a. The maximum number of clients is 32, since this is the number commonly supported by real devices such as the MGate MB3170/3270 as shown in its technical documentation [50]. Unlike the previous section, this experiment focuses on analyzing only the behavior of processing time for three-phases of mbapSSI, using the same tools as in the first experiment, i.e., expressions (4) and (5). For this purpose, 10 processing time measurements are collected for each of the three-phases.

#### 5.2.3. Performance of the Modbus Transaction Phase at n:1 Ratio

This experiment aims to measure the behavior of the latencies of the Modbus transaction phase under the stimulus of scaling the number of simultaneously connected Modbus clients in ratios of 4:1, 8:1, 16:1 and 32:1, following the architecture of Figure 4a. As in Section 5.2.1, this experiment uses the performance test illustrated in Figure 4b. Considering that mbapSSI Modbus frames are transacted over a secure channel, the experiment includes, as a benchmark, the same performance test applied to Modbus TCP.

#### 5.2.4. Measuring Transaction Throughput over the HFB Network

This experiment aims to determine the number of transactions per second that the different network architectures deployed are able to support, so that it is possible to analyze whether mbapSSI can cause overhead in the HFB network. For this purpose, 1000 transactions are issued in a variable range, for each of the networks (1:1, 4:1, 8:1, 16:1 and 32:1) defined based on Figure 4a. The Hyperledger Caliper tool [51], simplifies the transaction evaluation workflow. Thus, a comparison between five types of networks created from docker swarm clusters containing different architectures is carried out. Then, the maximum transaction throughput achieved will be compared with the number of interactions performed by clients and server with HFB, assuming that all devices are connected simultaneously. All nodes in the HFB-network deployed have the same participation in transaction endorsement.

## 6. Discussion of Results

### 6.1. Performance of mbapSSI Phases at 1:1 Ratio

This section contains the results of the experiment described in Section 5.2.1 and its main objective is to determine a set of benchmark metrics to be used as a starting point for the analysis in the other sections, since it constitutes the best case. Table 3 and Table 4 summarize the maximum and minimum values of the processing time, as well as the number of interactions with the blockchain for both the client and the server for each of three-phases of mbapSSI. The registration phase presents a longer processing time for S_H_ than the C_H_, since it not only creates and registers its identity but also creates and registers the DiD Document. The channel securing phase presents similar behavior for both entities, as Figure 3 illustrates, based on the same number of interactions of both entities (C_V_ and S_V_) with the blockchain, as well as a number of symmetric interactions between C_H_ and S_H_. It should be noted that only C_H_ participates in the verification phase. The processing time of the verification phase involves the generation of a JWT by C_H_, as well as a single interaction with the VDSR, hence it is as simple as C_H_’s processing time in the registration phase.

At this point, the performance test of the Figure 4b is performed on both the mbapSSI secure connection and insecure Modbus TCP connection, which is used for benchmark purposes. Table 5 collects the maximum and minimum latencies for both connections. Since the latencies achieved are below the time defined as typical reaction time for an industrial TCP connection, i.e., 100 ms [18], it can be considered that for this mbapSSI baseline case, the achieved latency values are acceptable.

### 6.2. Performance of the Three-Phases of mbapSSI Based on n:1 Ratio

In this section we analyze the effects, in terms of maximum and minimum processing times, when the number of mbapSSI client scales follows the next ratios: 4:1, 8:1, 16:1 and 32:1. In this way, a comparison with the results of Section 5.1 is established. Figure 5a shows the behavior of processing time when the number of clients deployed for the registration phase is increased. For each architecture, only one server is deployed and, since C_H_ and S_H_ do not interact with each other at this phase, the processing times obtained for the server are practically the same as those achieved in Section 5.1. In this way, the server processing time is closer to the processing time for registering 8 clients if the maximum and minimum processing times are considered. Figure 5b also shows the behavior of the processing time when the number of clients deployed for the securing channel phase is increased. Unlike the registration phase, in this phase there is a client–server interaction, hence the behavior of the server will depend on the number of clients simultaneously interacting with it. The results show that the higher the number of clients, the higher the maximum processing time for the server, which can be explained by the overhead accumulated after processing interactions with some clients. Despite this scenario, when comparing the extreme cases for the server, i.e., the reference architecture (1:1) and the worst case, i.e., the 32:1 architecture, the differences between the maximum and minimum processing time values are still acceptable: 45.8 ms for the maximum processing time and 32.7 ms for the minimum processing time. Figure 5c shows the behavior of the processing time for the verification phase when the number of clients increases. When comparing the minimum and maximum processing time values between clients of the reference architecture, i.e., 1:1, and the 32:1 architecture, the difference is 10.9 ms for the minimum processing time and 13.2 ms for the maximum processing time, keeping acceptable performance levels in both cases.

### 6.3. Performance of the Modbus Transaction Phase at n:1 Ratio

The isolation of the fourth phase of mbapSSI aims to ensure that once the first three phases of mbapSSI have been successfully completed, the Modbus transactions maintain the requirements of simplicity, compatibility, and interoperability. For this purpose, using as a reference the latency achieved by applying the performance test in Figure 4b over Modbus TCP, the same experience is compared over Modbus TLS. Figure 6 includes the repetition for the experiment (latencies’ measurements for Modbus transactions) for the next client–server architectures: 4:1, 8:1, 16:1 and 32:1.

The best- and worst-case scenarios are compared to reach conclusions, i.e., the latency of the reference architecture 1:1 and the worst case, i.e., the 32:1 architecture. Thus, 1 TCP client has a maximum latency of 0.54 ms, while 1 TLS client has a maximum latency of 1.37 ms. Using the same measurement criteria per client, 32 TCP clients have a latency of 39.3 ms, while 32 Modbus TLS have a latency of 63.3 ms. All these times achieved are below the time defined as typical reaction time for an industrial TCP connection, i.e., 100 ms [18], hence it can be considered that mbapSSI allows to keep the aforementioned Modbus properties.

### 6.4. Measuring Transaction Throughput over the HFB Network

As mentioned in Section 5.2.4, 1000 transactions are sent at variable rate for each of the five possible network scenarios to determine the performance of these transactions. Based on the chaincode design of Figure 2, the *setIdentity* method was selected as the test method because it is performed by the participant entities at boot time. Considering that the block size is fixed at 100 transactions per block or 2 MB [35] and that the validation policy forces all organizations to validate transactions, i.e., all organizations have the same weight in the consensus, Figure 7 shows the throughput behavior for each of the defined architectures.

The saturation point should be highlighted for each of the five cases, since it represents the point from which the number of transactions per second stops growing. Analyzing the extreme cases, 1:1 and 32:1, with the best and worst saturation point, respectively, the 1:1 architecture reaches saturation for a throughput of 129.6 tps, i.e., of the 150 transactions sent, 129.6 transactions are successfully committed into the ledger in 1 s, while the 32:1 architecture reaches saturation for a throughput of 37.2 tps, i.e., of the 50 transactions sent, 37.2 transactions are successfully committed into the ledger in 1 s. The remaining transactions sent will be committed in subsequent intervals. To determine the overhead that the mbapSSI boot time can cause on HFB, from Figure 5 the number of interactions are computed until all identities have been registered. Thus, for this interval, the identity registration of the C_H_ and the S_H_ will be considered as a concurrent transaction. Table 6 collects for each architecture the concurrent transactions, the saturation point, and the sending rate associated with that saturation. Hence, in the 32:1 scenario, where the simultaneous registration of the 32 clients and 1 server (which requires 66 concurrent transactions), reach the saturation point, the remaining pending transactions would be committed in the next second. However, considering that this registration is carried out only once and is not interfering in the transaction execution time, the boot-time overhead of mbapSSI over HFB might not be considered a problem.

Nevertheless, there are also ways to avoid this overhead. It is necessary on the one hand to decrease the block size, since performance analyses show decreasing the block size implies better throughput and lower latency [52]. On the other hand, reducing the number of endorsing peers in the HFB-network allow fewer entities to execute transactions and therefore transaction could be ordered into the block quickly; however, this implies that there must be greater trust between the organizations.

This results discussion section has demonstrated the feasibility of mbapSSI in terms of performance and scalability. The achievement of reasonable processing times in each of the phases that ensure Modbus transactions, the latency levels below the benchmark for industrial environments achieved in Modbus transactions and the optimal performance of mbapSSI interactions with HFB, for environments with up to 32 deployed organizations attest that statement.

## 7. Conclusions

Modbus is a widely used IIoT protocol based on three main features: simplicity, compatibility, and interoperability, but which lacks security. In this regard, Access Control Systems emerge as a solution. However, common access control solutions are based on centralized systems that include well known drawbacks: a single point of failure and limited scalability. This manuscript examines an Access Control System based on Self-Sovereign Identity over Hyperledger Fabric Blockchain from an Identity and Access Management perspective. The designed decentralized identity system supports on-chain authentication and authorization. Hence, it provides not only security for Modbus connections, but also ensures scalability in environments with more than one organization. The performed experiments and a subsequent critical discussion demonstrate that processing times achieved for the registration, channel securing and verification phases, as well as the latency achieved for Modbus transactions and the throughput achieved for mbapSSI transactions over HFB guarantee both the feasibility and scalability of mbapSSI and the simplicity, compatibility, and interoperability of Modbus. However, self-sovereign identity in machine-to-machine (IIoT) environments is in its infancy, therefore, our next approach is to eliminate the need for user involvement, achieving a fully machine-to-machine interaction, for which we are currently studying the OAuth machine-to-machine scheme to guarantee access to resources [42]. This approach should be fully compliant with DID standard [23]. Likewise, authorization mechanisms based on verifiable credentials are under study, a line in which Siemens is undertaking important steps to support selective disclosure based on Zero Knowledge Proofs [53]. These are the subjects of future lines of research. Additionally, mbapSSI supports the default accountability provided by blockchain, so we are also developing specific chaincodes for monitoring logs and event emission that integrate access control systems.

## Figures and Tables

**Figure 1 sensors-21-05438-f001:**
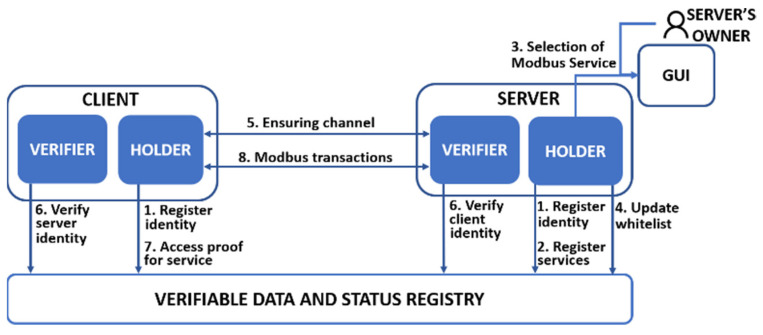
Overview of mbapSSI.

**Figure 2 sensors-21-05438-f002:**
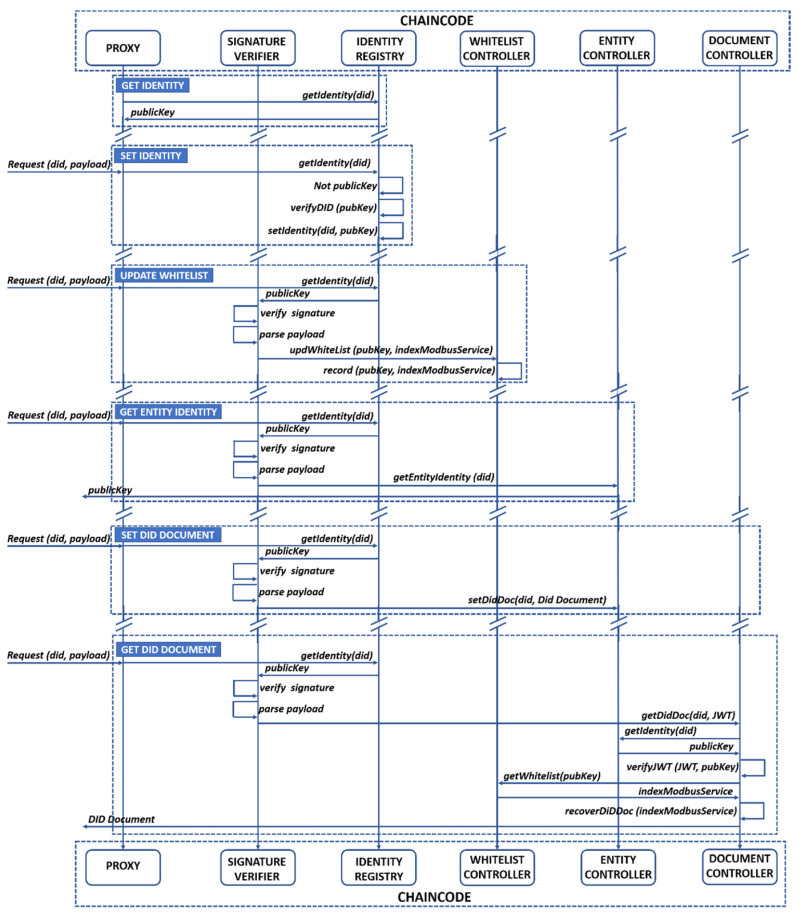
Chaincode design.

**Figure 3 sensors-21-05438-f003:**
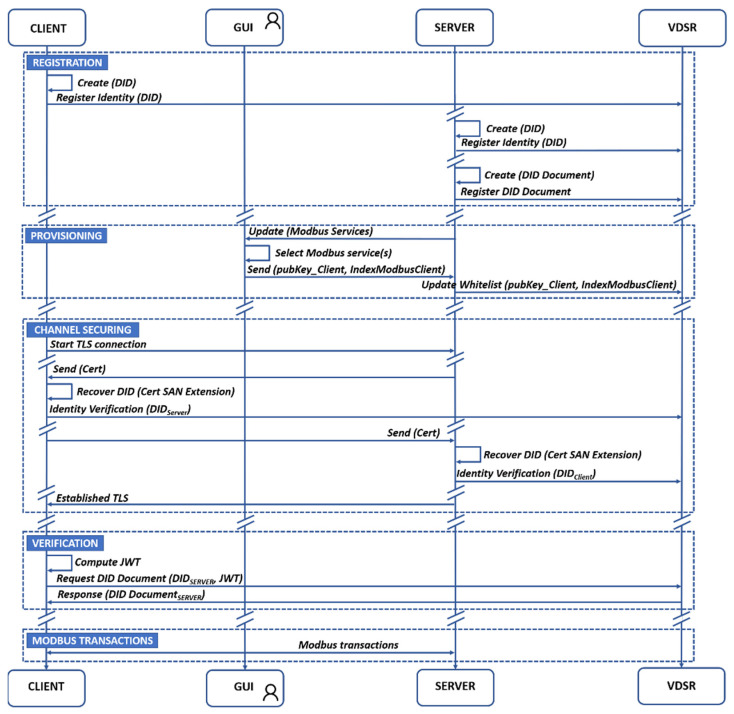
Interaction between entities and information processing by VDSR.

**Figure 4 sensors-21-05438-f004:**
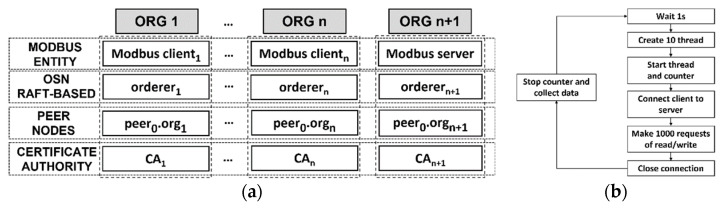
(**a**) Testbed applied on performance measure; (**b**) flow chart of Modbus performance test.

**Figure 5 sensors-21-05438-f005:**
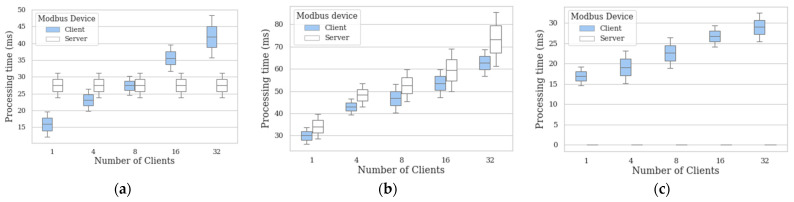
Processing time of (**a**) registration phase; (**b**) channel securing phase; (**c**) verification phase.

**Figure 6 sensors-21-05438-f006:**
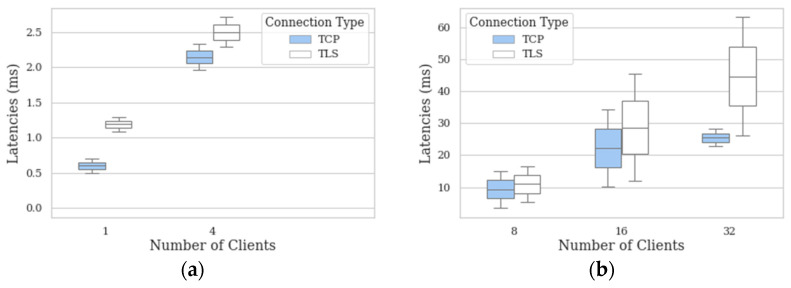
Latencies’ measurements in the fourth phase for the following architectures: (**a**) 1:1 and 4:1; (**b**) 8:1, 16:1 and 32:1.

**Figure 7 sensors-21-05438-f007:**
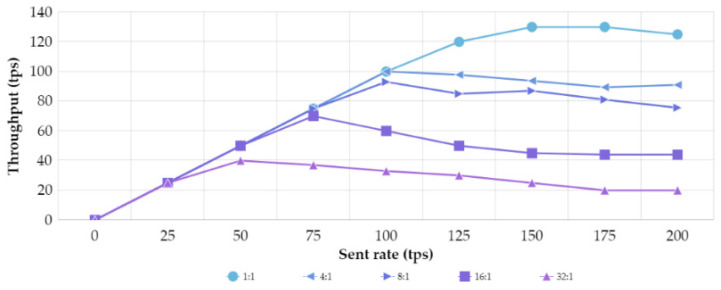
mbapSSI throughput behavior for different architectures.

**Table 1 sensors-21-05438-t001:** Participation of SSI parties on mbapSSI’s phases.

Phase	User	Holder	Verifier	VDSR
Registration		X		X
Provisioning	X			X
Channel securing		X	X	X
Verification		X		X
Modbus transaction		X		

**Table 2 sensors-21-05438-t002:** Modbus devices’ behaviors.

Phase	Device
Registration	C_H_, S_H_
Channel securing	C_H_, S_H_, C_V_, S_V_
Verification	C_H_
Modbus transaction	C_H_, S_H_

**Table 3 sensors-21-05438-t003:** Processing time measurements of three-phases of mbapSSI for the client.

Phase	Modbus Client Time Min	Modbus Client Time Max	HFB Invocations	HFB Queries
Registration	12.1 ms	19.7 ms	1	0
Channel Securing	26.2 ms	33.8 ms	0	1
Verification	14.6 ms	19.2 ms	0	1
Total	52.9 ms	71.7 ms	1	2

**Table 4 sensors-21-05438-t004:** Processing time measurements of three-phases of mbapSSI for the server.

Phase	Modbus Server Time Min	Modbus Server Time Max	HFB Invocations	HFB Queries
Registration	23.9 ms	31.2 ms	2	0
Channel Securing	28.7 ms	39.6 ms	0	1
Total	51.6 ms	70.8 ms	2	1

**Table 5 sensors-21-05438-t005:** Latencies measurements of the fourth phase of mbapSSI.

Phase	Client-Server TCP Min Time	Client-Server TCP Max Time	Client-Server TLS Min Time	Client-Server TLS Max Time
Modbus transaction	0.49 ms	0.54 ms	1.12 ms	1.37 ms

**Table 6 sensors-21-05438-t006:** Determination of the boot-time overhead of mbapSSI over HFB.

Architecture	mbapSSI Concurrent Transactions	Saturation Point (tps)	Sent Rate (tps)
1:1	2	129.6	150
4:1	10	100	100
8:1	18	97	100
16:1	34	70	75
32:1	66	37.2	50

## Data Availability

Not applicable.

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
