# Peer review of "Modbus Access Control System Based on SSI over Hyperledger Fabric Blockchain"

_sensors, 2021, doi:10.3390/s21165438_

Round 1

Reviewer 1 Report

Review Report

Manuscript

Modbus access control system based on SSI over Hyperledger Fabric Blockchain

Manuscript # 1299076

Submitted in the journal “Sensors

Summary

The manuscript’s contribution is the design and implementation of a Modbus Application Protocol Secured based on Self-Sovereign Identity (mbapSSI) over Hyperledger Fabric Blockchain (HFB) that ensures not only on-chain authentication and authorization but also the other phases of an access control system: identification, auditing, and account- ability. The mbapSSI system is also compliant with Decentralized Identifier (DID) standard.

Comments

  • This manuscript is well written and proportioned in every aspect, therefore, is acceptable to be publish.

Suggestions

  • Please check for English, Grammar, or typo errors.

Author Response

Please, see the attached document.

Reviewer 2 Report

The authors present an idea of the Modbus control system based on SSI concepts implemented on the Hyperledger Fabric blockchain. The general intention and idea seem fair, but the authors failed to outline why such a proposed system would make sense. The general problem of the paper is that it is probably not structured in the best direction. Thus the main Chapter (i.e., Chapter 3) is too much intertwined with “basic” SSI concepts. As a result, the Chapter's main objective, i.e., to present the design, features, novelty, etc., of the proposed system is not enough clearly described. Furthermore, the related work section should discuss related works and the relation of specified works with the article. Moreover, the performance evaluation is not set well enough to prove the feasibility of the proposed system (i.e., mbapSSI). For a more detailed list of remarks, please see below. In my opinion, the paper cannot be accepted in that form. The main parts (CH3 + some parts of CH4) of the paper should be re-written to expose the novelty of the proposed system features and not only “generic” features/characteristics of “any” SSI system.

  • Statement starting in line 34, i.e., “... These types of common solutions, based on centralized systems, are optimal in single-organization environments...” needs to be justified and/or supported by the appropriate reference.
  • The statement in line 73 is not correct. Blockchain technology could be used to achieve SSI paradigm goals, but it doesn't need to be even used (authors stated that BC is the core of SSI).
  • The statement in line 75 is not correct. The cited document is as defined in its content (see Disclaimer), prepared for the European Commission, and reflects the views only of the authors and not the EC.
  • Please clarify/describe how the statement starting in line 90 (including the Reference nr. 11) is relevant for this work.
  • Related works should discuss (i.e., describe why the selected works are “related”) the relevance (relation) with this work.
  • The chapter describing the background and “theory” behind SSI and its related concepts is missing. Some parts from Sections 3.1.1., 3.1.2., 3.1.3. could be moved to that “missing” chapter (to give general descriptions of concepts) and later "re-use" them in the "design and implementation” part of the article.
  • Roles discussed in Section 3.1.1. are in the cited W3C standard proposal specified as in the Verifiable Credential ecosystem. The correlation between Verifiable Credentials and the SSI paradigm should be discussed (preferably in the missing “background” chapter – see issue nr. 6).
  • Please describe, why the VDSR needs to be trusted? (see line nr. 137).
  • Would you please describe why the VDSR is the mediator to “create” relevant information, e.g., identifiers (see line nr. 137)? As is written now, it is against the SSI paradigm, where the parties themselves create identifiers.
  • Would you please justify why the statement about the trusted database (lines 138-140) is relevant for this work? (+ this statement validates that the statement in line 73 is not correct - see issue nr. 2).
  • Reference nr. 26 is not appropriate enough (i.e., slides of presentation) to be used in such an important design choice. + how the adopted approach is different from the approach described in reference nr. 25 (W3C Decentralized Identifiers (DIDs) v1.0 specification).
  • Content between lines 151-154 needs to be clarified and more widely discussed. i.e., the relation between DID and X.509 cert
  • The whole section about DID Method, and how the DID document is resolved is missing.
  • It is unclear, what is the role of “user” specified in Table 1.
  • The role of the Modbus Server (generally) needs to be described.
  • Chapter 3.2. should be written more clearly i.e., to extract all the “SSI” related concepts (DID document, Verifiable Credentials, etc) backgrounds into a dedicated chapter. Then this Chapter (i.e., 3.2) should clearly describe how these concepts are used in mbapSSI. A lot of important details are missed, and the relations between the aforementioned concepts are not clear (DID method, DID communication, how Verifiable Credentials (if?) are employed, and why they are or they are not, etc).
  • Relates to issue nr. 12 – what is an added value (and how, why) to create the system that incorporates DID’s and X.509 in the “permissioned setting” simultaneously. Please describe and elaborate differences between the “only X.509” and “DID+X.509” approach (within the HFB setting, that as default uses X.509 certificates for identification and authentication purposes).
  • Where the DID related private key is stored? And where X.509 related to private key?
  • What is the role of DID communication (where it is used – if it is), and how it (could) fit in this article? What is the relation between TLS communication (line 202) and DID communication? If DID communication is not used – why then introduce DID’s, if one of the main features of “DID” i.e., secure communication is not employed. And vice-versa, if DID communication is used – why then use Modbus TLS communication? All these questions remain unanswered during the manuscript.
  • The design of the mbapSSI is not sufficiently described – General comment.
  • The novelty i.e., the correlation between Modbus and “SSI concepts” are not sufficiently described, and are unclear. – General comment
  • Chapter 3.3. describes the implementation of a standard “DID” related system functionalities (SET DID, GET DID, etc). Would you please elaborate on the novelty from the system design perspective (i.e., not implementation)
  • General (i.e., missing background chapter) concepts are not properly explained (they are explained in the chapter that describes the design of the mbapSSI), and this results, that also the design choices, goals and features (first part of Chapter 3) of the mbapSSI are not enough clearly presented/justified. – General comment
  • Three performance metrics are evaluated, i.e., processing time, latency and throughput in “three-phases”, i.e., registration, channel securing, verification. The main issue of this evaluation is that the evaluation measures the performance of the HFB and not the mbapSSI (at least in the case of Registration and Verification, where none of the Modbus operations is executed). i.e., processing times, latencies, and throughput should be the same in the chosen deployment setting, independent of the mbapSSI system. It is expected that results should be the same for the XYZ use case, which is deployed in such an HFP network with such deployment settings. i.e., the results (at least the majority of them) are expected (for example – throughput - see https://doi.org/10.1007/s11280-021-00874-7)

Author Response

Please, see attached document.

Reviewer 3 Report

The paper addresses an interesting issue of  using SSI in Modbus. Similar to many IIoT systems Modbus is not designed with built in security. The authors propose using SSI for access control to Modbus services. The paper is well organized. There are some language errors, what could be expected from non native English speakers. Although language could be improved it does not prevent understanding of content.

The main issue this reviewer has with the paper is real need for SSI in Modbus environment.  I am not close with Modbus but would expect it to be deployed in close environments with central authority. For such environment there are more traditional security solutions. Such solutions are well know, tested and easier to implement.

The functioning of Modbus should have been better explained for readers of this issue. The simplicity, compatibility and interoperability of Modbus are mentioned several times without providing background why that is  a case.

Hyperledger Fabric Blockchain is mentioned and used in design. A reason for using HFB over other SSI implementations is not clear.

Design of mbapSSI is reasonably well explained. Although, from the reviewer's point of views the design is more complicated than necessary.

Test bed setup is not clear. Authors mention AWS EC2 instance. No explanation of Modbus environment.

Results confirm that processing times and transaction throughput are within acceptable limits for industrial environment. But these values are much (2 to 3 times) higher than without SSI.

Author Response

Please, see attached document.

Reviewer 4 Report

The article proposes a security solution for the Modbus IIoT (Industrial Internet of Things) protocol. The three main features that enable Modbus success are compatibility, simplicity, and interoperability. The main parts of the proposed Modbus Application Protocol Secured based on Self-Sovereign Identity (mbapSSI) system are SSI parties, Decentralized Identifier (DID), DID Document, and authorization whitelist. The implemented system is based on self-sovereign identity over hyperledger fabric blockchain compliant with a decentralized identifier. The designed decentralized identity system complies with the DID standard while supporting on-chain authentication and authorization. The system also ensures scalability in environments with more than one organization.

The article is well written, but a few commas and rephrasing could be checked for the text to be improved. The tables are well-positioned and have relevant titles, but they do not have the same format type (tables 1 and 2 use a different table format than the rest of the tables). Most of the diagrams have an appropriate size, but some should've been bigger and better centered on the paper's template. Figures should have uniform alignment in the paper. For example, figure 4 is aligned to the left while figure 1,2,3 are aligned to the right.

The article demands a knowledge base for the reader to fully understand the proposed solution for the security of the Modbus IIoT protocol.

The paper has a medium length and a good structure. The title is brief and gives a better understanding of the paper's content. The abstract terms are appropriately handled. The Abstract and the Introduction chapters include enough details about the article's subject matter. A few charts should be a little more extensive or use another color theme for better and quicker observation. The Conclusion chapter summarizes the paper's main ideas. At its end, the Author's Contributions, Funding, and Conflicts of Interest are well highlighted subchapters.

The authors should pay more attention to the grammar rules and the spelling check: “a Role based Access Control” - a Role-based Access Control “is structured as follow” - is structured as follows “as identity” - as an identity “local human machine” - local human-machine “is also associated to” - is also associated with “is a data structure which contains” - is a data structure that contains “provides the device with the decentralized identity” - provides the device with a decentralized identity

Most references are up to date, but more recent related work regarding practical use cases should be added, for example for critical infrastructures:

- Yalcinkaya, Erkan, et al. "Empowering ISA95 compliant traditional and smart manufacturing systems with the blockchain technology." Manufacturing review (2021).

- Sachian, Mari-Anais, et al. "Securing the smart grid: A blockchain-based secure smart energy system." 2019 54th International Universities Power Engineering Conference (UPEC). IEEE, 2019.

- Saha, Shammya Shananda, et al. "Integrating hardware security into a blockchain-based transactive energy platform." 2020 52nd North American Power Symposium (NAPS). IEEE, 2021.

Author Response

Please, see attached document.

Round 2

Reviewer 2 Report

The authors presented a revised version of a paper that adequately addresses all exposed issues of the manuscript's first version. Thus the article has been significantly improved. Despite, the performance evaluation has been improved and additionally discussed to verify the feasibility (performance and scalability) of the proposed system (i.e., mbapSSI), the statement (lines 54-55) that mbapSSI is compliant with DID standard is not supported (i.e., evaluated and proven) in the article. I encourage authors to provide a new subsection to elaborate/evaluate and thus demonstrate that mbapSSI is compliant with the standard (recommendation) specified in the article. The best-case scenario for this could be the usage of DID test suite (https://w3c.github.io/did-test-suite/) with which can be determined whether or not a given implementation is conformant with DID specification.

Author Response

Dear reviewer,

Thank you for your comment. Please see the attached document.
